# Persistence comparison of two Shiga-toxin producing *Escherichia coli* (STEC) serovars during long-term storage and thermal inactivation in various wheat flours

**Ian S. Hines**[1], **Tom Jurkiw**[1], **Emily Nguyen**[2], **Martine Ferguson**[1], **Sultana Solaiman**[1], **Elizabeth Reed**[1], **Maria Hoffmann**[1], **Jie Zheng**[1]*

**1** Center for Food Safety and Applied Nutrition, U.S. Food and Drug Administration, College Park, MD, United States of America, **2** Joint Institute of Food Safety and Applied Nutrition, College Park, MD, United States of America

* jie.zheng@fda.hhs.gov

## Abstract

Foodborne outbreaks associated with Shiga toxin-producing *Escherichia coli* (STEC) contaminated wheat flour have been an increasing food safety concern in recent decades. However, there is little literature aimed at investigating the impact of different flour types on the persistence of STEC during storage and thermal inactivation. Therefore, two serovars of STEC, O121 and O157, were selected to inoculate each of five different types of common wheat flours: whole wheat, bleached, unbleached, bread, and self-rising. Inoculated flours were examined for the stability of STEC during storage for up to 42 days at room temperature (RT) and $a_w$ ~0.56. Additionally, the thermal resistance of O121 and O157 under isothermal conditions at 60, 70, 80, and 90˚C was analyzed for the inoculated flours. STEC storage persistence at RT was generally not affected by flour type, however, decreases of 1.2 and 2.4 log CFU/day within whole wheat flour for O121 and O157, respectively, were significantly lower than other flours. Though few differences were identified in relation to flour type, O121 exhibited significantly better survival rates than O157 during both equilibrium and storage periods. Compared to an approximate 6 log reduction in the population of O157, O121 population levels were reduced by a significantly lower amount (~3 log) during the entire storage period at RT. At each isothermal temperature, the impact of flour type on the thermal resistance capabilities of O121 or O157 was not a significant factor and resulted in similar survival curves regardless of serovar. Instead of exhibiting linear survival curves, both O121 and O157 displayed nonlinear curves with some shoulder/tail effect. Similar for both O121 and O157, the predicted decimal reduction time (D-value) decreased from approximately 25 min to around 8 min as the isothermal temperature increased from 60˚C to 90˚C. Results reported here can contribute to risk assessment models concerning contamination of STEC in wheat flour and add to our understanding of the impacts of flour type and STEC serovar on desiccation stability during storage and isothermal inactivation during thermal treatment.

**Data Availability Statement:** All relevant data are within the manuscript and its Supporting Information files.

**Funding:** The author(s) received no specific funding for this work.

**Competing interests:** The authors have declared that no competing interests exist.

## Introduction

Wheat crops are a global commodity with 2022 worldwide production amounting to 0.8 billion tonnes [1]. Processing of wheat kernels into consumer-ready flour involves a multistep process wherein microbial contamination can occur at many different points. Due to its low water activity ($a_w < 0.85$), raw flour had been considered a microbiologically safe product by preventing bacterial proliferation [2,3]. However, flour can serve as a vector for microbiological contaminants including foodborne pathogens such as *Salmonella enterica* and *Escherichia coli* [4–6]. These bacterial pathogens have been the cause of several outbreaks including hundreds of cases associated with consumption of raw or undercooked flour products that can also have economic ramifications [7]. One such product, ready-to-bake cookie dough, was implicated as the source of a Shiga toxin-producing *E. coli* (STEC) outbreak in 2009, resulting in 77 clinical cases and the recall of 3.6 million packages [6]. There have continued to be several US-based STEC-associated outbreaks from flour consumption, with the latest having occurred in 2021 [8–10]. Since the first reports of flour-associated outbreaks, there has been an increased effort to investigate the cause of disease transmission and to identify potential prevention strategies.

Consumer-ready whole wheat and regular wheat flours typically have a shelf life of 3–9 months and 9–15 months, respectively, during which time potentially pathogenic bacteria can persist [11]. Though flour matrices do not allow for proliferation, STEC can survive in the low moisture environment for several months and up to years past the shelf life, albeit at a level below standard plate enumeration [12,13]. However, this low population level can still be an infectious dose with the consumption of raw or undercooked flour products. Additionally, desiccation stress may provide cross-protection from external forces such as heat treatments [4,14]. Pre- and post-milling treatment options for reducing the microbial load in flours prior to retail purchase can include the use of tempering agents, heat (both dry and steam), irradiation, and ozone [15–17]. Unfortunately, treatment efficacy can vary [18–20], and some treatments may result in decreased product quality. For example, addition of acidified water to flour can effectively limit microbial growth and survival, but its addition may negatively impact the sensory quality of the flour [20]. This is especially important when analyzing the makeup of different types of flour, but it is not currently known what effect, if any, the varying compositions of different flours have on the survival of microbial pathogens known to persist in flour. Dry heating of flour products remains a viable path toward reducing microbial loads while maintaining flour quality, however, the industrial practice times and temperatures vary greatly [16].

The present study sought to examine the fates of STEC serovars in different flour types up to 42 days of storage at a controlled water activity ($a_w$) level. In addition, the thermal resistances of two outbreak-associated STEC serovars in different wheat flour types at several isothermal temperatures were evaluated.

## Materials and methods

### Wheat flours

Flours used for this study included all-purpose bleached, all-purpose unbleached, whole wheat, bread, and self-rising (Gold Medal, General Mills, Minneapolis, MN). Samples from all five different flours were sent to Silliker Inc.–Merieux NutriSciences (Crete, IL) for nutritional chemistry analyses. The following panels were included for analysis: amino acids complete, fat by fatty acid profile, ICP MS full mineral screen, organic acids, and sugars.

## Bacterial strains and culturing

Two strains of Shiga toxin-producing *Escherichia coli* (STEC) were obtained from the stock culture collection of the Division of Microbiology, Center for Food Safety and Applied Nutrition, U.S. Food and Drug Administration, College Park, MD. STEC O121:H19 (strain CFSAN051458) was originally isolated from an outbreak-associated all-purpose flour in 2016, and STEC O157:H7 (strain EC1734) was isolated from an outbreak-associated cookie dough in 2009. Both strains were stored in brain heart infusion (BHI) broth containing 25% glycerol at -80˚C. Frozen cells were streaked onto trypticase-soy agar (TSA) and incubated at 35˚C overnight to obtain individual colonies. Individual colonies were transferred to 5 mL trypticase-soy broth (TSB) and incubated at 35˚C overnight. To obtain a high cell concentration, 100 μL of the overnight culture was spread onto each of three TSA plates prior to overnight incubation at 35˚C. Formed lawns were resuspended with 1 mL sterile 0.1% buffered peptone water (BPW) per each plate, and 1 mL of the resulting suspension was used for the subsequent flour inoculations.

## Inoculated wheat flour preparation and equilibration

Samples of all flour types were stored at room temperature (RT) within a humidity-controlled glove box maintained at 50% relative humidity for preconditioning at least 3–4 days before each inoculation. Glove box humidity was maintained by a relative humidity controller (Microcontroller Model 5100, Electro-Tech Systems, Inc., Perkasie, PA) set to 50%. The controller was connected to two Air Cadet® vacuum pressure pumps (Cole-Parmer, Vernon Hills, IL) to control the flux between two canisters containing sterile dH$_2$O and silica gel beads, respectively.

One-hundred grams of flour were inoculated with 1 mL of the bacterial suspension within a 1-gallon Ziploc bag (S. C. Johnson, Racine, WI) and thoroughly massaged by hand for at least 5 min until there were no visible clumps. Inoculated flours were placed back into the humidity-controlled glove box at RT and the inoculum was allowed to fully dry in the flour overnight. After equilibration for 24 hr, the inoculated flour was sifted through a sterile aluminum sifter in a powder hood (Class II Type A2 biosafety cabinet) to remove any observable clumps. Any remaining clumps were further ground using a sterile mortar and sterile pestle. Homogeneity of inoculum was evaluated by randomly removing six 1 g samples from an inoculated batch of flour (100 g) after the equilibration period. Each sample was resuspended with 9 mL sterile 0.1% BPW within a 1.627 L filtered Whirl-Pak bag (Nasco, Fort Atkinson, WI) and mixed by hand for 2 min. Flour resuspensions were then serially diluted within sterile 1X phosphate buffered saline (PBS) and spread onto TSA and an *E. coli*-selective chromogenic agar: cefixime tellurite sorbitol-MacConkey (CT-SMAC) for O157; and non-O157 STEC plating medium (R & F products, Downers Grove, IL) for O121, in duplicate. The plates were incubated at 35 ± 2˚C for 18–24 h and enumerated to verify the homogeneity of inoculum distribution.

## Water activity measurements

The water activity of the flour samples prior to inoculation, during homogeneity testing, and on each sampling day was measured at ~22˚C with an AquaLab 4TE Dew Point Water Activity Meter (Decagon Devices Inc., Pullman, WA, USA) following the manufacturer's instructions. For each measurement, 1 g of inoculated flour was removed from each bag at the time of testing. Storage study water activities were measured in duplicate.

## Storage study

Each inoculated batch of flour (100 g) was placed in a humidity-controlled glove box at RT (~21˚C) with 50% relative humidity. Flour samples were taken 0- and 2-days post-inoculation followed by weekly sampling up to 42 days. After reaching the detection limit by direct plating (3 log CFU/g), samples were enriched before detection. Enrichment was accomplished by resuspending two 5 g flour samples in 45 ml of 1X modified BPW with pyruvate (mBPWp) and massaged within Whirl-Pak bags. Resuspended flour samples were statically incubated at 37˚C for 5h prior to addition of Acriflavin-Cefsulodin-Vancomycin (ACV) supplement per FDA's Bacteriological Analytical Manual (BAM) (Feng et al., 2020). The samples were then incubated at 42 ± 1˚C statically overnight (18-24h). BAM Chapter 4a [21] was then followed for detection/isolation of STEC from overnight enrichments. Sampling stopped after two consecutive negative results were obtained.

## Isothermal treatments

Isothermal studies were performed using an oven (Thermo Scientific, Waltham, MA) preset to 3˚C above the desired isothermal treatment temperature (60˚C, 70˚C, 80˚C, or 90˚C) for at least 30 min prior to each experiment. The 3˚C increase was used to compensate for the temperature drop observed during sample transfer. Autoclave-sterilized aluminum test canisters were filled with inoculated flour (1 g, ~3 mm depth) and quickly transferred into the oven. The temperature was then set back to the testing temperature. For each strain, the following time/temperature combinations were tested: treatment times of 30, 60, 90, and 120 min at 60˚C; 15, 30, 45, and 60 min at 70˚C; 5, 10, 15, and 20 min at 80˚C; 5, 7, and 10 min at 90˚C. Two samples per each strain were subsequently removed at each time point. After being removed from the oven, treated flour-containing canisters were immediately put on ice for at least 5 min to stop thermal inactivation prior to enumeration. Positive controls (time = 0), not subjected to thermal treatment, were also sampled in duplicate. The $a_w$ of inoculated flour was measured immediately prior to isothermal treatment. All thermal inactivation experiments at each temperature were performed in duplicate from freshly inoculated flour batches.

## STEC recovery and enumeration

To enumerate surviving STEC populations during storage, bags containing inoculated flour were removed from storage in the glove box. Triplicate 2 g samples from each bag were mixed with 18 ml 0.1% BPW each and massaged by hand for 2 min before serial dilution and plating as described above for homogeneity testing.

After thermal treatment, flour samples (1 g/sample) were aseptically transferred from the test canisters to sterile Whirl-Pak bags and mixed with 9 ml 0.1% BPW. After hand massaging for 2 min, samples were serially diluted, and spread plated as described above for homogeneity testing.

## Statistical analyses and parametric regression modeling

Storage trial data for each of the three samples was subjected to a one-way analysis of variance (ANOVA) test on the slopes of the log reduction of *E. coli* during the equilibrium period (0–2 days post inoculation (dpi)) and the storage period (2-29/42 dpi). The one-way ANOVA calculated the effect of flour type on the ability of STEC to persist during equilibrium and storage. A Tukey's Honest Significant Differences post-hoc test was run after the ANOVA to indicate any pair-wise differences between flour types. A Student's T-test was used to test for significant differences in equilibrium and storage period slopes between each serovar.

Variance components (VC) for replicate plating, sample, and sampling week were estimated using a multi-level mixed effect analysis of variance, after taking into account variability arising from flour, sampling time, strain, and plating differences. VC's were estimated using the function anovaMM in the R package VCA [22] with replicate plating nested within sample and sample nested within sampling week.

Due to the small sample size, the thermal inactivation data was analyzed using Bayesian methods. The relationship between flour type and isothermal inactivation was assessed at each temperature using a Bayesian first-order isothermal inactivation model [23].

$$log_{10}N_t = log_{10}N0 - t/\delta \qquad (1)$$

where $N_t$ is the cell count (colony forming units (CFU)/g flour) at time t (min), N0 is the cell count at time 0, and δ is the D-value (min). The D-value represents the time required for reduction of the bacterial count by a factor of 10 at the specified temperature. This model was run separately for each serovar and plating medium.

Models (1) were fit using PROC MCMC in SAS 9.4 (SAS Institute, Cary NC) with an assumed gamma (shape = 10, scale = 1) prior distribution for the D-value. Two thousand posterior samples drawn using the Metropolis algorithm and the posterior distribution of the D-value was summarized using 89% credible intervals [24], with non-overlapping intervals indicating differences between flour types.

## Results

### Chemical profiling of different flour types

Complete analytical results from the five different types of wheat flours are shown in S1 Table in S1 File. Compared to other flour types (Fig 1A and 1C), whole wheat flour had a higher content of amino acids (12.59%, *w/w*) and cis-monounsaturated fatty acids (18%, *w/w*). In addition, whole wheat flour had ~3 fold higher concentration of sucrose and ~2 fold higher concentration of glucose than the other flours (Fig 1B). Whole wheat flour was also rich in potassium and magnesium. Maltose was the most abundant sugar observed in all flours, with the highest concentration found in all-purpose unbleached flour (9.08%, *w/w*) (Fig 1B). While concentrations of minerals including Phosphorus (P), Sodium (Na), Calcium (Ca), and Aluminum (Al) were higher in self-rising flour than the other flours, Potassium (K) and Magnesium (Mg) were more abundant in whole wheat flour (Fig 1D).

### Background flora and water activity (a_w) of various wheat flours

The background flora enumerated via aerobic plate count was near the limit of detection 3 log CFU/g) for all five flours. All flour samples used for the storage and thermal inactivation studies had an $a_w$ of 0.55 ± 0.028 pre-inoculation. Water activity during the storage study for all five flours inoculated with each STEC strain had a slight increase post inoculation (S1 Fig in S1 File). After the equilibrium period, the $a_w$ of flours post inoculation was 0.59 ± 0.024 for O121 strain-inoculated flours and 0.56 ± 0.016 for O157 strain-inoculated flours and maintained throughout the entire storage study. In flour samples used for the thermal inactivation study, the $a_w$ measured prior to thermal treatments was 0.54 ± 0.048 for O121-inoculated flours and 0.56 ± 0.033 for O157-inoculated flours.

### STEC survival in various wheat flours during storage

The storage study for STEC O121 in different wheat flours was carried out for 42 days at RT and sampled weekly after the first two days post inoculation (Fig 2A). In general, the levels of

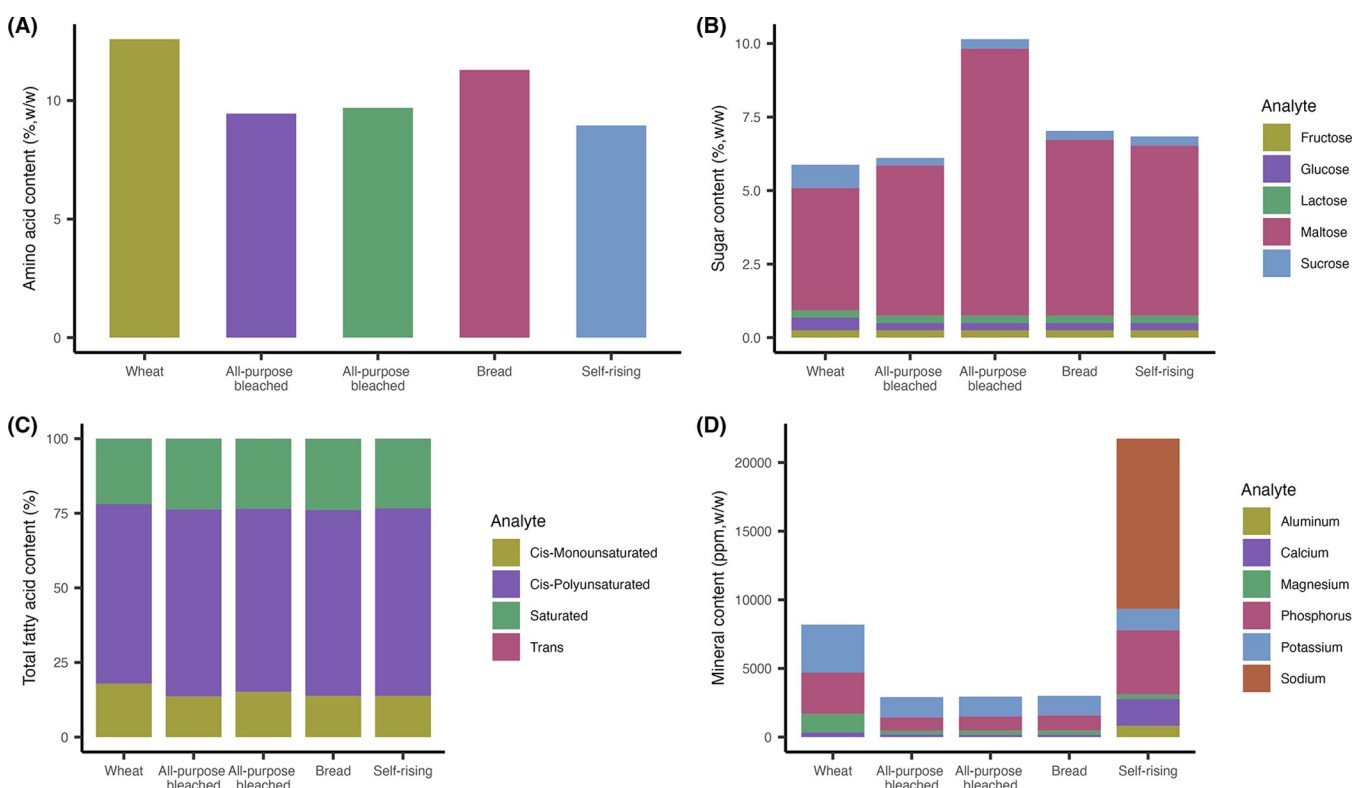

**Fig 1. Analyte profiles of five different wheat flours.** Samples of various wheat flours were analyzed for their analyte profiles. Analytes measured included (A) amino acids, (B) sugars, (C) fatty acids, (D) and minerals.

O121 had an initial reduction of ~1 log CFU across all flour types for the first 2 days (equilibrium period) followed by an additional ~2 log CFU reduction in O121 population across all flour types for the remaining 40 days of storage. The reduction rates of O121 slowed down from an average of -0.61 log CFU/day in the first two days of the equilibrium period to an average of -0.043 log CFU/day until day 42 of storage across all flour types. A total of ~3 log CFU reduction was observed by the end of the study for all flours. During the equilibrium period, no significant differences in the reduction rate (i.e., slope) were detected between the flours. After the conditioning period, however, the reduction rate was significantly different among several wheat flours during storage. Specifically, a significantly ($P < 0.05$) lower reduction rate of the O121 population was observed in bleached flour (-0.037 log CFU/day) than in bread flour (-0.05 log CFU/day). Similarly, the O121 population in whole wheat flour during storage following equilibrium decreased at a significantly ($P < 0.05$) lower rate (-0.033 log CFU/day) than in bread (-0.05 log CFU/day), self-rising (-0.047 log CFU/day), and unbleached (-0.046 log CFU/day) flours, respectively.

In comparison to O121, O157 could only be enumerated from inoculated wheat flours for 29 days (Fig 2B). After days 29 of storage, enumeration of O157 was below the enumerable detection limit, however, its presence was confirmed following enrichment through 42 days. The reduction rates of the O157 populations during both equilibrium and storage periods were significantly higher than those of the O121 populations ($P < 0.001$). O157 population levels decreased by an average of 3 log CFU (-1.49 log CFU/day) during the equilibrium period followed by a further reduction of ~3 log CFU during the storage period (-0.11 log CFU/day) for a total loss of ~6 log CFU across all types of flours. In particular, the O157 population

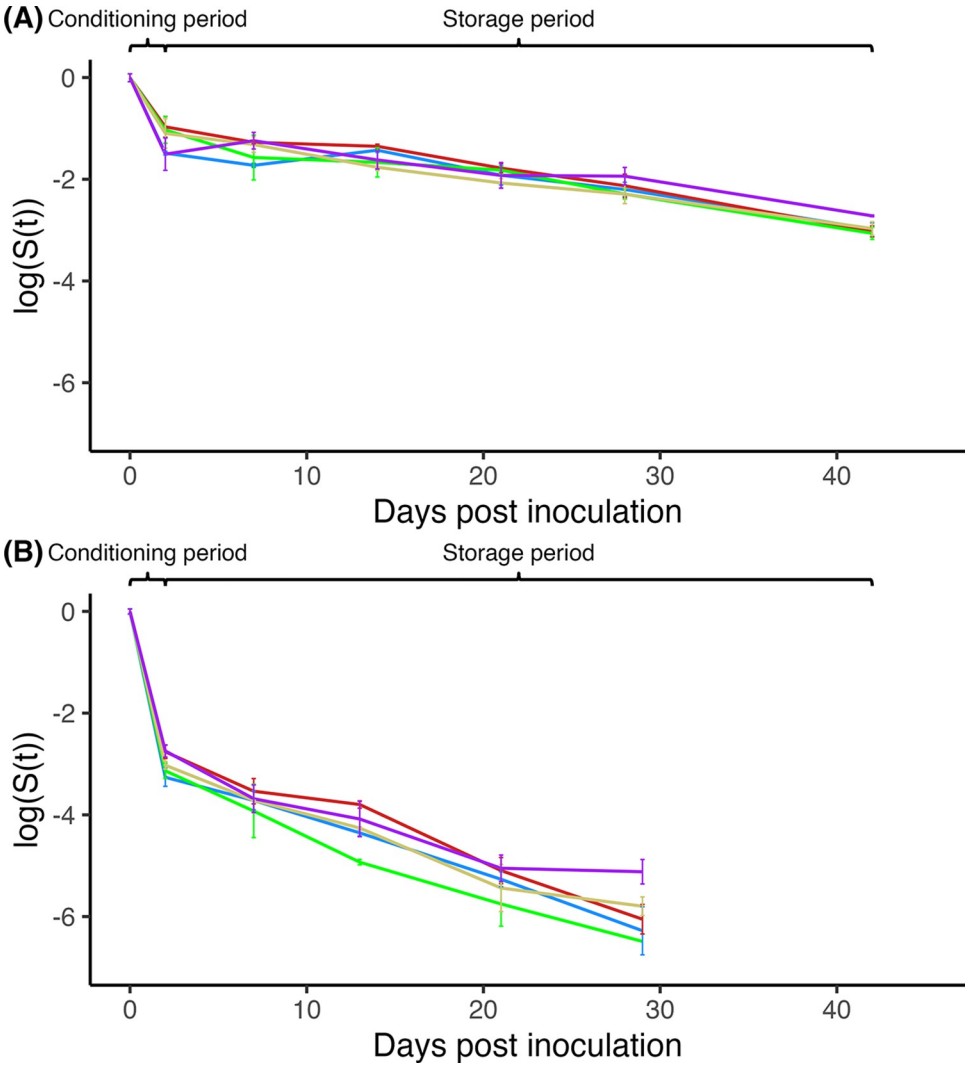

**Fig 2. STEC survival in different wheat flours during long-term storage.** The following wheat flour types were used: Whole wheat (purple), all-purpose bleached (blue), all-purpose unbleached (yellow), bread (red), and self-rising (green). Three samples of each flour, inoculated with either (A) STEC O121 or (B) STEC O157, were sampled and enumerated. The log ratio of culture levels (logS(t), wherein $S(t) = N_i/N_0$) is plotted against the time. Error represents the mean ± standard deviation of three samples.

within bleached flour decreased at a significantly ($P < 0.05$) higher rate (-1.6 log CFU/day) than in the whole wheat flour (-1.4 log CFU/day) and bread flour (-1.4 log CFU/day), respectively during the equilibrium period. It was also observed that, during the storage period, the O157 population within whole wheat flour decreased at a significantly ($P < 0.05$) lower rate (-0.088 log CFU/day) than in self-rising flour (-0.128 log CFU/day).

## Isothermal inactivation kinetics of STEC in wheat flours

As the temperature increased from 60°C to 90°C, the time needed to reduce the pathogen population decreased. Therefore, the intervals between sampling timepoints also decreased with increased temperature. The O121 and O157 population levels measured before isothermal treatment were used as the initial populations for determination of log-population survivor

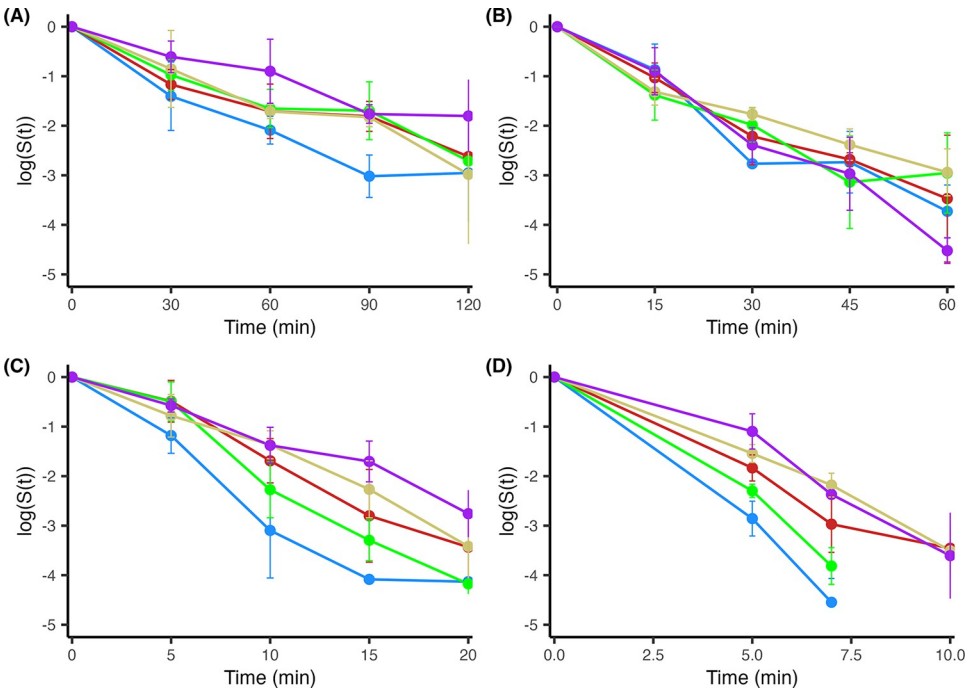

**Fig 3. STEC O121 survival in different wheat flours following isothermal treatments.** The following wheat flour types were used: Whole wheat (purple), all-purpose bleached (blue), all-purpose unbleached (yellow), bread (red), and self-rising (green). Two independently inoculated samples of each flour were sampled in duplicate and subjected to isothermal treatments at (A) 60°C, (B) 70°C, (C) 80°C, and (D) 90°C. The log ratio of culture levels (logS(t), wherein S (t) = $N_i/N_0$) is plotted against the time. The length of time at the specified isothermal temperature was determined for each temperature individually. Error represents the mean ± standard deviation of two independent samples with two replicate samplings each (n = 4). Points without error bars indicate only a single sample above the limit of detection. Lines cut off when the culture goes below countable numbers.

ratios. Overall, both O121 and O157 STEC populations in any type of wheat flour appeared to decline much faster during the earliest stages of isothermal inactivation treatment, however, the reduction rate appeared to slow down during the later incubation time points (Figs 3 and 4). It was also noted that the isothermal inactivation curves of O121 exhibited minor, albeit not significant, variation between the different flours for each of the temperatures tested (Fig 3).

Following equilibration for 48 h (equilibrium period), there was a 3-log drop in the initial O157 population which was on average ~1 log CFU lower than the initial O121 population in wheat flours also as evidenced in the storage survival section. Isothermal inactivation kinetics of O157 in different types of flour also exhibited some degree of variation although not significant. No consistent flour-specific curves were observed in the thermal resistance capabilities of O157 at each different temperature, which was similarly observed with O121-inoculated flours (Fig 4).

## Model D-value determination for *E. coli* O121 and O157

The isothermal inactivation curves of *E. coli* O121 (Fig 3) and O157 (Fig 4) generally followed a nonlinear decreasing trend. A Bayesian first-order isothermal inactivation model (Eq 1) was used to identify any significant differences in the time to first decimal reduction (D-value) between the different flours at each isothermal condition for each serotype tested (Table 1). Variance attributed to plating replicates within samples (0.24% of total variance) and sample replicates (~0% of total variance) was relatively small compared to the inter-experiment

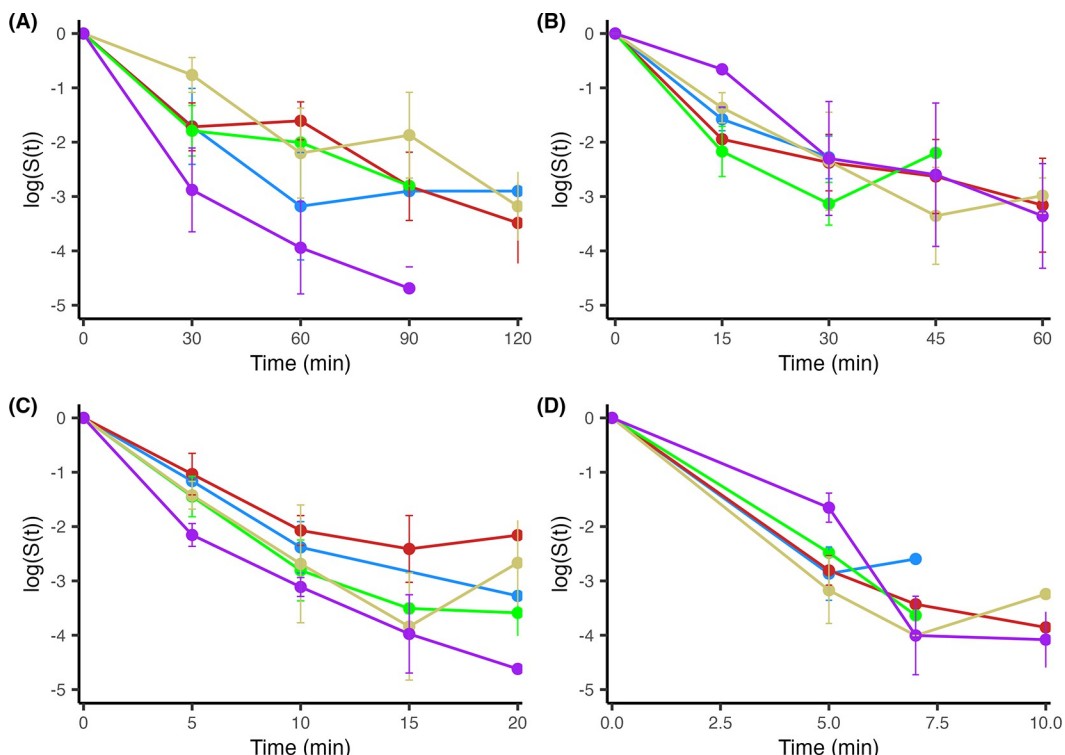

**Fig 4. STEC O157 survival in different wheat flours following isothermal treatments.** The following wheat flour types were used: Whole wheat (purple), all-purpose bleached (blue), all-purpose unbleached (yellow), bread (red), and self-rising (green). Two independently inoculated samples of each flour were sampled in duplicate and subjected to isothermal treatment at (A) 60°C, (B) 70°C, (C) 80°C, and (D) 90°C. The log ratio of culture levels (logS(t), wherein $S(t) = N_i/N_0$) is plotted against the time. The length of time at the specified isothermal temperature was determined for each temperature individually. Error represents the mean ± standard deviation of two independent samples with two replicate samplings each (n = 4). Lines cut off when the culture goes below countable numbers.

(week) variance (~1.1% of total variance). Therefore, cell counts per dilution within a sample and between samples per time/temperature (sample replicates) were averaged together within each independent experiment as indicated by "week" for statistical analyses. Posterior median D-values including their associated 89% credible intervals were generally similar across the flour types and temperatures for both organisms, apart from O157-inoculated whole wheat and bread flours subjected to the 60°C heating treatment and enumerated with the CT-SMAC chromogenic medium. For each organism, the average D-values were: ~25, 16, 8, and 8 min for 60, 70, 80, and 90°C, respectively.

## Discussion

Foodborne outbreaks associated with the consumption of wheat flour have become an area of concern for food safety experts following several high-profile outbreaks in the past couple of decades. The low moisture content of these foods prevents growth of foodborne pathogens such as *Salmonella* and *E. coli*, however, these organisms are still able to persist and can cause disease if consumed raw or undercooked. Despite investigations regarding the association between these pathogens and the flour matrix, very few studies have sought to look at the effect of flour type on bacterial persistence.

The present study examined the impact of different wheat flour varieties on the desiccation tolerance and thermal resistance of *E. coli* serovars. Commonly used wheat flours, including

**Table 1. D-value estimates as calculated by Bayesian first-order isothermal inactivation kinetic model of different STEC-inoculated wheat flours.**

| Strain | Medium | Flour[1] | Posterior mean D-value \| 89% credible interval (min) | | | |
|---|---|---|---|---|---|---|
| | | | 60°C | 70°C | 80°C | 90°C |
| O121 | R&F® | Whole wheat | 26.7 \| 21.0–35.2 | 13.8 \| 11.5–18.4 | 10.7 \| 7.3–15.6 | 6.8 \| 4.3–13.5 |
| | | AP-bleached | 26.8 \| 20.9–34.8 | 14.3 \| 11.4–17.4 | 7.7 \| 5.0–12.0 | 7.7 \| 3.6–11.9 |
| | | AP-unbleached | 26.1 \| 20.2–32.9 | 16.4 \| 12.4–21.5 | 7.9 \| 7.0–9.0 | 6.9 \| 3.8–12.1 |
| | | Bread | 23.1 \| 18.2–29.8 | 14.3 \| 10.8–18.0 | 7.0 \| 5.5–11.0 | 6.3 \| 3.2–10.1 |
| | | Self-Rising | 23.8 \| 18.8–28.8 | 15.2 \| 11.4–19.1 | 5.6 \| 4.3–9.8 | 9.3 \| 4.5–14.2 |
| | TSA | Whole wheat | 26.9 \| 21.8–33.7 | 15.4 \| 12.2–20.9 | 10.4 \| 7.8–14.4 | 9.7 \| 5.4–15 |
| | | AP-bleached | 26.6 \| 20.6–33.3 | 14.4 \| 10.9–17.5 | 6.4 \| 5.0–9.3 | 5.6 \| 2.6–11.4 |
| | | AP-unbleached | 26.6 \| 21.7–33.0 | 18.4 \| 13.4–23.1 | 8.5 \| 7.2–11.0 | 8.1 \| 4.4–13.6 |
| | | Bread | 22.7 \| 16.7–31.2 | 15.8 \| 12.4–20.6 | 8.3 \| 6.4–12.5 | 6.9 \| 3.7–12.2 |
| | | Self-Rising | 22.9 \| 17.2–30.3 | 16.3 \| 12.5–20.8 | 7.5 \| 5.5–13.0 | 8.3 \| 4.0–13.6 |
| O157 | CT-SMAC | Whole wheat | 19.6 \| 16.1–23.6 | 16.2 \| 12.1–21.2 | 6.9 \| 5.0–10.5 | 7.7 \| 5.2–13.0 |
| | | AP-bleached | 22.2 \| 17.2–28.3 | 14.5 \| 10.5–20.4 | 8.3 \| 5.8–13.0 | 8.4 \| 4.8–13.3 |
| | | AP-unbleached | 22.9 \| 17.2–29.4 | 16.0 \| 12.8–21.4 | 7.1 \| 5.1–11.9 | 6.3 \| 3.0–12.3 |
| | | Bread | 29.4 \| 24.9–35.0 | 15.1 \| 12.3–18.8 | 8.2 \| 5.5–11.9 | 7.7 \| 3.6–15.3 |
| | | Self-Rising | 20.3 \| 15.4–26.4 | 11.4 \| 8.0–15.8 | 6.4 \| 4.5–9.9 | 8.5 \| 4.7–13.7 |
| | TSA | Whole wheat | 24.4 \| 17.7–31.1 | 18.0 \| 12.5–24.0 | 11.2 \| 8.2–15.6 | 7.7 \| 4.9–12.5 |
| | | AP-bleached | 22.3 \| 16.3–28.5 | 16.7 \| 13.2–20.6 | 8.4 \| 6.3–12.9 | 8.1 \| 4.1–13.7 |
| | | AP-unbleached | 23.8 \| 17.4–31.7 | 15.8 \| 11.6–21.3 | 8.1 \| 5.8–12.4 | 6.8 \| 3.1–13.1 |
| | | Bread | 31.5 \| 23.6–38.8 | 17.2 \| 12.6–21.1 | 9.2 \| 6.7–12.8 | 7.1 \| 3.6–12.4 |
| | | Self-Rising | 20.6 \| 16.0–25.8 | 15.6 \| 12.0–19.9 | 6.6 \| 4.9–11.5 | 8.2 \| 4.3–14.3 |

[1] AP = All-purpose.

whole wheat, bleached, unbleached, bread, and self-rising, were used in this study. Despite some compositional differences in amino acid, sugar, and mineral contents, most flour types did not significantly (P > 0.05) alter the ability for STEC to survive in wheat flour. However, both O121 and O157 serovars appeared to persist better in whole wheat flour than the other flours at RT with 50% relative humidity. Considering the water activity was similar between the five wheat flours, some specific flour components of the whole wheat flour may be the most likely contributing factor. For instance, the higher presence of amino acids in whole wheat flour may enable better desiccation tolerance [25]. Further testing on the influence of specific compositional differences is warranted to gain a better understanding of what mechanisms may allow for differential STEC desiccation persistence.

The persistence of STEC in wheat flour was more significantly influenced by the specific serovar than by the composition of wheat flour. Interestingly, at RT and $a_w$ of ~0.55, O121 was able to persist longer and generally at a higher population level than O157 for the duration of the study. At similar $a_w$ and temperature, Forghani *et al* [12,26], however, reported O157 had a slower decline than O121 up to 12 weeks of storage. Strain variation and inoculation method [27] may contribute to the differences observed between storage studies. Expanding to other organisms, STEC persistence during storage in wheat flours is generally worse than that of *Salmonella* [26,28,29]. Some bacteria that are more tolerant to desiccation also appear to become more resistant to oxidative protein damage during drying [30]. Between STEC serogroups, O157:H7 cells exhibit differential expression of oxidative stress response genes, such as *stx*, compared to other serovars such as O104:H4, O145:NM, O111:NM, O26:H11, and O103:H2 [31,32], though little literature exists for STEC gene expression in low moisture foods such as

wheat flour. The molecular mechanisms involved in enhanced tolerance of some pathogens during desiccation are still not fully understood.

It was found that the flour type did not significantly ($P > 0.05$) alter STEC's thermal resistance capability at any of the temperatures investigated here. In fact, despite the lower survival of O157 compared to O121 post equilibration after inoculation, both organisms followed similar thermal inactivation kinetics in the same types of wheat flour at $a_w$ of ~0.54 with time to first decimal reduction values (D-values) calculated as ~25, 16, 8, and 8 min for 60, 70, 80, and 90˚C, respectively. Increasing the temperature to 80˚C or higher did not substantially increase the efficacy of isothermal inactivation, as similar D-values were observed at both 80˚C and 90˚C. Varied D-values for STEC in wheat flour under the same temperatures were observed in other isothermal inactivation studies [12,26,33,34]. Factors including serovar, strain, $a_w$, inoculation level and method, and model used to simulate the thermal inactivation survival curve all impact the estimation of the D-value. In the present study, while O121 was shown to have better desiccation tolerance during storage at RT, no significant ($P > 0.05$) difference was found between O157 and O121 in isothermal resistance when treated at 60, 70, 80 and 90˚C, respectively. Forghani *et al* [12,26] also reported that the thermal kinetics were very similar among all the STEC serogroups tested (O45, O121, O145, O26, O103, O111 and O157) when treated at 60 and 70˚C in the studies. However, much lower logarithmic reduction time for O121 was found when compared to other serovars such as O26 and O45 for isothermal treatment at 60˚C [33].

In summary, the ability for STEC to resist desiccation in wheat flour during storage and tolerate isothermal treatment was not significantly affected by the composition of the tested wheat flours. Small differences were observed with whole wheat flour, indicating that there may be some potential for similarly composed wheat flours to provide better persistence for STEC. Desiccation tolerance of STEC in wheat flour during storage at RT is serovar dependent, however, the logarithmic reduction time (D-value) of STEC in wheat flour was not influenced by serovar during isothermal treatments. This lack of significant difference in D-value between different serovars indicates thermal treatment remains an effective method for mitigating risk of STEC in wheat flour. Higher temperatures with shorter treatment times may be desirable to achieve multiple log reduction in STEC populations in wheat flour since heat treatments at or just below 100˚C do not appear to severely impact flour structural integrity [35]. Multiple factors need to be considered for estimation of logarithmic reduction time to better predict flour thermal process lethality. The findings of the present study further expand the knowledge on STEC risk assessment of wheat flour.

## Supporting information

**S1 File. Supplementary file.**
(DOCX)

## Author Contributions

**Conceptualization:** Ian S. Hines, Tom Jurkiw, Maria Hoffmann, Jie Zheng.

**Data curation:** Ian S. Hines, Tom Jurkiw, Emily Nguyen, Martine Ferguson, Sultana Solaiman, Elizabeth Reed, Jie Zheng.

**Formal analysis:** Ian S. Hines, Tom Jurkiw, Emily Nguyen, Sultana Solaiman, Elizabeth Reed, Jie Zheng.

**Investigation:** Ian S. Hines, Tom Jurkiw, Emily Nguyen, Elizabeth Reed, Jie Zheng.

**Methodology:** Ian S. Hines, Tom Jurkiw, Martine Ferguson, Maria Hoffmann, Jie Zheng.

**Project administration:** Ian S. Hines, Tom Jurkiw, Maria Hoffmann, Jie Zheng.

**Resources:** Ian S. Hines.

**Supervision:** Ian S. Hines, Tom Jurkiw, Maria Hoffmann, Jie Zheng.

**Validation:** Ian S. Hines, Tom Jurkiw, Martine Ferguson, Jie Zheng.

**Visualization:** Ian S. Hines, Martine Ferguson, Elizabeth Reed.

**Writing – original draft:** Ian S. Hines, Martine Ferguson, Jie Zheng.

**Writing – review & editing:** Ian S. Hines, Tom Jurkiw, Martine Ferguson, Elizabeth Reed, Maria Hoffmann, Jie Zheng.

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
