## [Decision Letter · Decision Letter 0]

2 Jan 2024

PONE-D-23-32459Persistence comparison of two Shiga-toxin producing Escherichia coli (STEC) serovars during long-term storage and thermal inactivation in various wheat floursPLOS ONE

Dear Dr. Zheng,

Thank you for submitting your manuscript to PLOS ONE. After careful consideration, we feel that it has merit but does not fully meet PLOS ONE’s publication criteria as it currently stands. Therefore, we invite you to submit a revised version of the manuscript that addresses the points raised during the review process.

Use appropriate controls to demonstrate the temperature effect on E. coli without the flour==============================

We look forward to receiving your revised manuscript.

Kind regards,

Arun K. Bhunia, Ph.D.

Academic Editor

PLOS ONE

Additional Editor Comments:

Two reviewers rendered a mixed decision. One was favorable, while the second reviewer questioned the experimental design, especially the missing controls. Given such diverse opinions, the manuscript cannot be considered for publication. However, if the authors are willing to revise the manuscript as suggested, the manuscript could receive further consideration.

Reviewers' comments:

Reviewer's Responses to Questions

**Comments to the Author**

1. Is the manuscript technically sound, and do the data support the conclusions?

Reviewer #1: Yes

Reviewer #2: No

2. Has the statistical analysis been performed appropriately and rigorously? 

Reviewer #1: Yes

Reviewer #2: Yes

3. Have the authors made all data underlying the findings in their manuscript fully available?

Reviewer #1: Yes

Reviewer #2: Yes

4. Is the manuscript presented in an intelligible fashion and written in standard English?

Reviewer #1: Yes

Reviewer #2: Yes

5. Review Comments to the Author

Reviewer #1: Thank you for the nice work done. The manuscript is very well written. While I do not see much novelty compared to what we already knew about the topic, some of the side information as well as additional data presented adding up to the field convinced me that having it published is better than not!

Reviewer #2: In the present study titled “Persistence comparison of two Shiga-toxin producing Escherichia coli (STEC) serovars during long-term storage and thermal inactivation in various wheat flours” researchers tried to investigate the impact of different flour types on the persistence of STEC during desiccation and thermal inactivation. They have used two serovars of STEC, O121 and O157, to inoculate each of five different types of common wheat flours: whole wheat, bleached, unbleached, bread, and self-rising. They have examined for the stability of STEC during storage up to 42 days under isothermal conditions at 60, 70, 80, and 90oC. These researchers found that STEC storage persistence was generally not affected by flour type, however, two different strains survived differently, O121 survived significantly better than O157 for both conditioning and storage periods. However, without using appropriate controls in this study, it seems difficult to draw any conclusion.

Major comments:

1. For the effect of isothermal temperatures on both serovars of STEC (Fig 3 and Fig 4), researchers didn’t use appropriate controls. They should use an “only temperature control”, bacteria growing in water/PBS without any flour type to check if the effects observed were due to change in only temperature parameter. Then, the effect of temperature on serovars can be subtracted from the observed effects of temperatures on serovars in the different flour types. Without this control it is irrelevant to draw any conclusion.

2. Quality and labelling of figures are very poor.

6. PLOS authors have the option to publish the peer review history of their article (what does this mean?). If published, this will include your full peer review and any attached files.

Reviewer #1: **Yes: **Fereidoun Forghani

Reviewer #2: **Yes: **Shivendra Tenguria

---

## [Author Response · Author response to Decision Letter 0]

12 Feb 2024

Reviewer #1: Thank you for the nice work done. The manuscript is very well written. While I do not see much novelty compared to what we already knew about the topic, some of the side information as well as additional data presented adding up to the field convinced me that having it published is better than not!

Response: We thank the reviewer for this very supportive comment!

Reviewer #2: In the present study titled “Persistence comparison of two Shiga-toxin producing Escherichia coli (STEC) serovars during long-term storage and thermal inactivation in various wheat flours” researchers tried to investigate the impact of different flour types on the persistence of STEC during desiccation and thermal inactivation. They have used two serovars of STEC, O121 and O157, to inoculate each of five different types of common wheat flours: whole wheat, bleached, unbleached, bread, and self-rising. They have examined for the stability of STEC during storage up to 42 days under isothermal conditions at 60, 70, 80, and 90oC. These researchers found that STEC storage persistence was generally not affected by flour type, however, two different strains survived differently, O121 survived significantly better than O157 for both conditioning and storage periods. However, without using appropriate controls in this study, it seems difficult to draw any conclusion.

Response: We appreciate the reviewer's comment. As highlighted by the reviewer, our objective is to incorporate new information regarding the influence of various flour types on the persistence of STEC during storage at RT and following thermal inactivation. In our analyses, we employed time “0” at storage temperature (i.e., RT) to assess STEC persistence in flour during storage or time “0” right before thermal treatment as controls to investigate STEC survival in flour during thermal inactivation. In the Bayesian first-order isothermal inactivation model we used in the study, D-value was calculated for each serovar at each isothermal temperature. Comparison of D-values between the serotypes was carried out under the same isothermal temperature for the same type of wheat flour. No statistical difference was found in the thermal resistance curves between the two serovars at each isothermal temperature. We believe we have implemented appropriate controls to derive meaningful conclusions.

Major comments:

1. For the effect of isothermal temperatures on both serovars of STEC (Fig 3 and Fig 4), researchers didn’t use appropriate controls. They should use an “only temperature control”, bacteria growing in water/PBS without any flour type to check if the effects observed were due to change in only temperature parameter. Then, the effect of temperature on serovars can be subtracted from the observed effects of temperatures on serovars in the different flour types. Without this control it is irrelevant to draw any conclusion.

Response: We thank the reviewer for the comment. In the investigation of thermal inactivation, as illustrated in Fig 3 and Fig 4, we have concluded that there is no significant impact of flour type on STEC survival under thermal inactivation, and there is no significant difference in survival between the serovars under the same temperature within the same flours. We employed time "0" in flour right before thermal treatment as a control, which is appropriate for this study, as the curves were log reductions relative to the 0-minute timepoint. As we compared the survival rate (D-value) of different serovars under the same thermal condition within the same flour, temperature is not a variable in the comparison. In addition, we tested the effects of different isothermal temperatures on the two serovars within only PBS as the reviewer suggested. Similar for both O157 and O121, after the initial 6-7 log reduction, no enumerable culture was recovered by the second timepoint for each isothermal temperature, indicating that the thermal treatment is very effective in killing pathogens without the protection of the food matrix. Since the goal of this study was to investigate the difference in survival of STEC serovars in multiple flour types, we believe that the more appropriate control is the time “0” sampling point. We apologize for any confusion and have adjusted the text to enhance the clarity.

2. Quality and labelling of figures are very poor.

Response: Each figure has been updated with a higher resolution, larger font size, and larger lines/dots within the plot. All manuscript figures have also been modified by PACE.

---

## [Decision Letter · Decision Letter 1]

19 Feb 2024

Persistence comparison of two Shiga-toxin producing Escherichia coli (STEC) serovars during long-term storage and thermal inactivation in various wheat flours

PONE-D-23-32459R1

Dear Dr. Zheng,

We’re pleased to inform you that your manuscript has been judged scientifically suitable for publication and will be formally accepted for publication once it meets all outstanding technical requirements.

Kind regards,

Arun K. Bhunia, Ph.D.

Academic Editor

PLOS ONE

Additional Editor Comments (optional):

Thank you for addressing all the concerns raised by the reviewers. The article meets the journal standards for publication.

Reviewers' comments:

Reviewer's Responses to Questions

**Comments to the Author**

1. If the authors have adequately addressed your comments raised in a previous round of review and you feel that this manuscript is now acceptable for publication, you may indicate that here to bypass the “Comments to the Author” section, enter your conflict of interest statement in the “Confidential to Editor” section, and submit your "Accept" recommendation.

Reviewer #2: (No Response)

2. Is the manuscript technically sound, and do the data support the conclusions?

Reviewer #2: Yes

3. Has the statistical analysis been performed appropriately and rigorously? 

Reviewer #2: Yes

4. Have the authors made all data underlying the findings in their manuscript fully available?

Reviewer #2: Yes

5. Is the manuscript presented in an intelligible fashion and written in standard English?

Reviewer #2: Yes

6. Review Comments to the Author

Reviewer #2: Response: The authors have addressed all of my review comments.

7. PLOS authors have the option to publish the peer review history of their article (what does this mean?). If published, this will include your full peer review and any attached files.

Reviewer #2: No

---

## [Editor Report · Acceptance letter]

29 Feb 2024

PONE-D-23-32459R1 

PLOS ONE

Dear Dr. Zheng, 

I'm pleased to inform you that your manuscript has been deemed suitable for publication in PLOS ONE. Congratulations! Your manuscript is now being handed over to our production team.

Kind regards, 

on behalf of

Dr. Arun K. Bhunia 

Academic Editor

PLOS ONE